

# Probing non-thermal density fluctuations
# in the one-dimensional Bose gas

**Jacopo De Nardis$^{1\star}$, Miłosz Panfil$^{2}$, Andrea Gambassi$^{3}$,**
**Leticia F. Cugliandolo$^{4}$, Robert Konik$^{5}$ and Laura Foini$^{6}$**

**1** Département de Physique, Ecole Normale Supérieure, PSL Research University, CNRS,
24 rue Lhomond, 75005 Paris, France.
**2** Institute of Theoretical Physics, University of Warsaw,
ul. Pasteura 5, 02-093 Warsaw, Poland.
**3** SISSA — International School for Advanced Studies and INFN,
via Bonomea 265, 34136 Trieste, Italy.
**4** Sorbonne Universités, Université Pierre et Marie Curie — Paris 6,
Laboratoire de Physique Théorique et Hautes Energies,
4, Place Jussieu, 75252 Paris Cedex 05, France.
**5** CMPMS Division, Brookhaven National Laboratory,
Building 734, Upton, New York 11973, USA.
**6** Laboratoire de Physique Statistique, Département de l'ENS, École Normale Supérieure,
PSL Research University, Université Paris Diderot, Sorbonne Paris Cité, Sorbonne Universités,
UPMC Univ. Paris 06, CNRS, 75005 Paris, France.

$\star$ jacopo.de.nardis@phys.ens.fr

## Abstract

**Quantum integrable models display a rich variety of non-thermal excited states with unusual properties. The most common way to probe them is by performing a quantum quench, i.e., by letting a many-body initial state unitarily evolve with an integrable Hamiltonian. At late times these systems are locally described by a generalized Gibbs ensemble with as many effective temperatures as their local conserved quantities. The experimental measurement of this macroscopic number of temperatures remains elusive. Here we show that they can be obtained for the Bose gas in one spatial dimension by probing the dynamical structure factor of the system after the quench and by employing a generalized fluctuation-dissipation theorem that we provide. Our procedure allows us to completely reconstruct the stationary state of a quantum integrable system from state-of-the-art experimental observations.**


# 1 Introduction

Motivated by the impressive experimental progress in engineering and manipulating cold atomic gases, the dynamics of isolated quantum many-body systems has recently been the subject of very intense theoretical [1–3] and experimental [4–9] investigations.

In particular, it has been firmly established that the nature of the eventual statistical description of the local properties of these systems in their stationary states depends crucially on them being integrable or not. In the former case, the presence of an extensive amount of (quasi-) local quantities $Q_n$ which are conserved by the unitary dynamics with Hamiltonian $H$, i.e., $[H, Q_n] = 0$, makes the system relax locally towards the generalized Gibbs ensemble (GGE) with density operator $\rho_{\text{GGE}}$ [10–25]. The GGE also holds at intermediate times in the case of a weak integrability breaking term [4, 9, 26–31]. In the non-integrable case, instead, the relaxation generically occurs towards the Gibbs ensemble (GE) $\rho_{\text{GE}}$, controlled solely by $H$ [29, 32, 33]. Both ensembles are characterized by certain parameters which are essentially determined by the expectation values of the conserved charges.

In the Gibbs ensemble, the only relevant parameter is the temperature $\beta^{-1}$ (and, possibly, the chemical potential $\mu$ in the grand canonical ensemble), as it completely determines the statistical and thermodynamic properties of the ensemble $\rho_{\text{GE}}$. Accordingly, it is very important to be able to measure the temperature. This can be done, for example, by weakly coupling the system with a thermometer or, alternatively, by studying the relationship between the measurable fluctuations occurring in the system and its response to external perturbations. The second approach relies on the so-called fluctuation-dissipation theorem (see, for example,

Refs. [34–36]).

In the generalized Gibbs ensemble $\rho_{\text{GGE}}$, the number of parameters playing the role of "generalized temperatures" which are necessary to characterize it completely is, in principle, extensively large: the task of fixing all of them would therefore appear impractical. However, it has been recently shown [13, 37, 38] that for a number of systems which are represented by *non-interacting* integrable models, it is possible to identify some "natural" observables, the correlations and responses of which can be used to determine the complete set of generalized temperatures *via* the corresponding fluctuation-dissipation ratios. In the presence of interactions, the analysis of integrable models becomes significantly more complex and it is therefore unclear, *a priori*, whether this approach carries over to interacting integrable models. In this work we show that this is actually the case, as the fluctuation-dissipation ratios (or, equivalently, the dynamic structure factor of the relevant fluctuations) can be used to infer all the parameters which determine $\rho_{\text{GGE}}$. In particular, we focus on the one-dimensional Bose gas described by the Lieb-Liniger model [39, 40] and we use the dynamical structure factor (DSF) $S(k, \omega)$, recently studied in Refs. [41, 42], in order to characterize $\rho_{\text{GGE}}$. This approach is remarkably useful as it provides a viable and concrete way of measuring $\rho_{\text{GGE}}$ in experiments with cold atoms in which integrability breaking terms can be controlled and made small [4, 9, 43, 44]. Moreover, we show that the fluctuation-dissipation ratio is an optimal tool to check the extent to which a system is thermalized in any experimental setting, even when the underlying model is not integrable.

To pursue this approach we formulate the micro-canonical version of the generalized Gibbs ensemble, that is, instead of a statistical operator $\rho_{\text{GGE}}$ we consider a single eigenstate $|\vartheta_{\text{GGE}}\rangle$ such that in the thermodynamic limit

$$\text{tr}[\rho_{\text{GGE}}O] = \langle \vartheta_{\text{GGE}}|O|\vartheta_{\text{GGE}}\rangle, \tag{1}$$

where $O$ is a local observable. This is fully analogous to the case of thermal equilibrium, where the eigenstate thermalization hypothesis [32, 33] allows one to compute expectation values at thermal equilibrium for a generic quantum state in terms of a single eigenstate.

Integrable models are generically characterized by *interacting* quasi-particles, i.e., by stable collective thermodynamic excitation modes which, due to their interactions, undergo elastic (non-diffractive) scattering. Therefore an energy eigenstate $|\vartheta_{\text{GGE}}\rangle$ can be specified by a corresponding mode occupation number $\vartheta(\lambda)$ of these excitation modes. Focusing on the case where only one type of excitation is present (as is the case in the one-dimensional Bose gas) each eigenstate of the many-body Hamiltonian $H$ can be formally specified by a single macroscopic mode occupation number $\vartheta(\lambda) \in [0, 1]$, defined as the fraction of the possible modes actually occupied within the rapidity interval $[\lambda, \lambda + d\lambda)$ of infinitesimal width $d\lambda$.

At thermal equilibrium within the (grand canonical) Gibbs ensemble $\rho_{\text{GE}} = e^{-\beta(H-\mu N)}$ with temperature $\beta^{-1}$ and chemical potential $\mu$, this function $\vartheta(\lambda)$ is given by [45]

$$\vartheta(\lambda) = \vartheta_{\text{GE}}(\lambda) \equiv \frac{1}{1 + e^{\beta(\omega(\lambda)-\mu)}}, \tag{2}$$

i.e., by the Fermi distribution in terms of the excitation energy $\omega(\lambda)$ (c.f. Eq. (12)) of the mode $\lambda$. On the other hand it is known that the dynamic structure factor:

$$S(k, \omega) = \int_{-\infty}^{+\infty} dt \int dx \, e^{i(kx-\omega t)} \Big[ \langle \rho(x, t)\rho(0, 0) \rangle - \langle \rho(x, t) \rangle \langle \rho(0, 0) \rangle \Big], \tag{3}$$

with $\rho(x, t)$ the density operator measuring the number of particles at position $x$ and time $t$, obeys the fluctuation-dissipation relation (detailed balance)

$$S(k, -\omega) = e^{-\beta \omega} S(k, \omega). \tag{4}$$

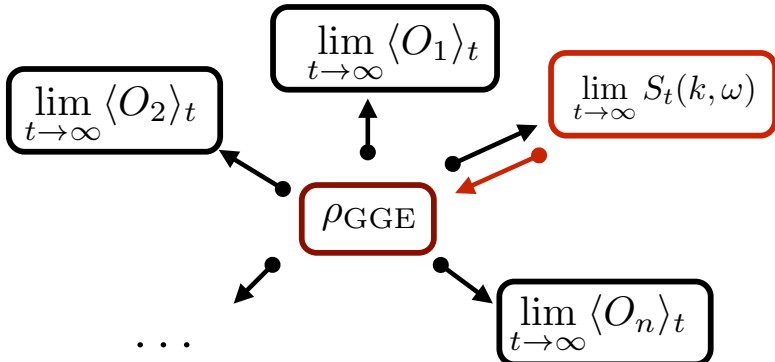

Figure 1: A GGE steady state $\rho_{\text{GGE}}$ determines (black arrows) the expectation value of any local operator $O_i$ in the long time limit and, in particular, also the dynamical structure factor $\lim_{t\to\infty} S_t(k,\omega)$. In this work we show that, given the dynamical structure factor in the long time limit after the quench, one can use its behavior at small momentum $k \to 0$ to fully reconstruct the GGE steady state (reversed red arrow) and, thus, the expectation value of any other local operator in the same stationary state.

Therefore knowing the dynamic structure factor we can determine $\beta$ and the corresponding mode occupation number $\vartheta_{\text{GE}}(\lambda)$.

In the case of non-thermal equilibrium with the stationary state given by a generalized Gibbs ensemble the mode distribution describing the GGE eigenstate (1) can be expressed as a thermal-like one (2) $\vartheta_{\text{GE}}(\lambda) \to \vartheta_{\text{GGE}}(\lambda)$ with generalized temperatures for each mode, namely with the substitution $\beta \to \beta(\lambda)$. It has recently been observed that a generalized detailed balance relation holds for such a state in the low momentum regime:

$$S(k,-\omega) = e^{-\mathscr{F}(k,\omega)}S(k,\omega) + \mathscr{O}(k^2), \qquad (5)$$

where $S(k,\omega)$ is evaluated on the post-quench stationary state and $\mathscr{F}(k,\omega)$ is a known functional of the mode distribution number. The corrections $\mathscr{O}(k^2)$ break the proportionality between the two DSF for a generic state as there is no detailed balance in a generic GGE state at any $k$. The functional $\mathscr{F}(k,\omega)$ is invertible and therefore from $S(k,\omega)$ we can infer $\vartheta_{\text{GGE}}(\lambda)$, or equivalently $\beta(\lambda)$, similarly to the thermal case, following a similar approach to the one introduced in Ref. [38]. From the function $\vartheta_{\text{GGE}}(\lambda)$ we can then determine any other expectation value of local operators in the long time limit analogously to what was done in Refs [19,46–48], see Fig. 1. This is the main idea behind our work.

We here focus on the one-dimensional Bose gas (i.e., on the so-called Lieb-Liniger model), for which the expression of the DSF $S(k,\omega)$ at low momenta and for a generic macroscopic state $|\vartheta\rangle$ has been recently calculated analytically in Ref. [41]. However our approach can be extended easily to any integrable model with one single type of quasi-particle in its spectrum. Indeed it was recently shown [49, 50] that given the DSF of an operator which is globally conserved, as for example the density $\rho(x)$ in the one-dimensional Bose gas (as the total density $n = \langle \int \rho(x)dx \rangle$ is conserved by the Lieb-Liniger Hamiltonian (6)), relation (5) holds generically for any model as a consequence of the generalized hydrodynamics theory [51,52].

The manuscript is organized as follows. In Secs. 2 and 3 we focus on the one dimensional Bose gas as described by the Lieb-Liniger model with no confining potential and we detail how our approach applies to this problem. Specifically, in Sec. 3 we focus on the case of an interaction quench from the BEC initial state. Finally, in Sec. 4 we present conclusions and perspectives. Some details of the analysis are reported in Appendix A and Appendix B.

In Appendix C we discuss an alternative approach to determining the GGE, which is valid in the strongly interacting regime of the Lieb-Liniger model. In Appendix D and Appendix E we discuss possible experimental issues in implementing the proposed protocol and how to cope with them. In Appendix F we comment on the ABACUS [53] software package that we used for the numerical evaluation of the dynamical structure factor.

## 2 From the dynamic structure factor of the one-dimensional Bose gas to the steady state

### 2.1 The Lieb-Liniger model

The Hamiltonian $H$ of a Bose gas consisting of $N$ particles in one spatial dimension (Lieb-Liniger model) [39,40] is

$$H = -\sum_{i=1}^{N} \partial_{x_i}^2 + 2c \sum_{i>j}^{N} \delta(x_i - x_j),$$

(6)

where $x_i$ denotes the position of the $i$-th particle, $c$ measures the strength of their repulsive contact interaction, and the units of measure are chosen such that $\hbar^2/2m = 1$. The model can be realized experimentally on atom chips and optical lattices, and in the past years many of its equilibrium properties have been studied in great detail [43, 44, 54–56]. We consider the gas to be confined within a finite segment of length $L$, with periodic boundary conditions (we consider the effect of a possible harmonic trap in Appendix D). The effective interaction is characterized by the dimensionless parameter $\gamma = c/n$, where $n = N/L$ is the linear particle density of the gas. In the thermodynamic limit $L \to \infty$ with fixed $n$ (or, alternatively with fixed chemical potential $\mu$), the eigenstates of $H$, referred to as Bethe states, are labeled by the filling function $\vartheta(\lambda) \in [0,1]$ which is the fraction of occupied modes within the rapidity interval $[\lambda, \lambda + d\lambda)$. Once the filling function $\vartheta(\lambda)$ is specified, the relevant thermodynamic quantities which characterize the state of the system can be easily determined [45]. For example, the total particle density $n$ and the energy density $e$ can be expressed in terms of the filling function and the total density of modes $\rho_t(\lambda)$, namely the Jacobian induced by the change of variable $k \to \lambda(k)$ with $k$ the real momentum of the particles, as

$$n = \int_{-\infty}^{+\infty} d\lambda \, \vartheta(\lambda)\rho_t(\lambda),$$

(7)

and

$$e = \int_{-\infty}^{+\infty} d\lambda \, \lambda^2 \, \vartheta(\lambda)\rho_t(\lambda),$$

(8)

respectively. In the case $c = \infty$ of hard-core bosons, the maximal number $2\pi\rho_t(\lambda)d\lambda$ of allowed modes within the interval $[\lambda, \lambda + d\lambda)$ is identically equal to 1. When $c$ is finite $\rho_t(\lambda)$ becomes a non-trivial function as the inter-particle interactions affect the modes, which are no longer homogeneously distributed in the space of rapidities. The total density $\rho_t(\lambda)$ of possible states and their actual occupation $\vartheta(\lambda)\rho_t(\lambda)$ are related by an integral equation [45]

$$2\pi\rho_t(\lambda) = \frac{dk(\lambda)}{d\lambda} = 1 + \int_{-\infty}^{+\infty} d\lambda' \, K(\lambda - \lambda')\vartheta(\lambda')\rho_t(\lambda'),$$

(9)

where the so-called scattering kernel is given by

$$K(\lambda) = \frac{2c}{\lambda^2 + c^2},$$

(10)

and the momentum $k(\lambda)$ is given in Eq. (11).

## 2.2 Excitations of a macroscopic state

In order to study correlation functions, it is important to understand the structure of the excitations of a macroscopic state $|\vartheta\rangle$ of the system which is specified by a certain occupation function $\vartheta(\lambda)$, see Refs. [41,42]. If the system is confined within a finite "volume" $L$, a single particle excitation corresponds to increasing by one the occupation number of a certain mode with quasi-momentum $p$. The resulting momentum $k(p)$ and excitation energy $\omega(p)$ can be expressed, in the thermodynamic limit, as

$$k(p) = p - \int_{-\infty}^{+\infty} d\lambda \, \vartheta(\lambda) F(\lambda|p), \tag{11}$$

$$\omega(p) = p^2 - 2 \int_{-\infty}^{+\infty} d\lambda \, \lambda \vartheta(\lambda) F(\lambda|p), \tag{12}$$

where we introduced the so-called back-flow function $F(\lambda|p)$ which describes the effects of the excitation on the occupation of the remaining modes, as schematically shown in Fig. 2. This function turns out to satisfy [45]

$$2\pi F(\lambda|p) = \theta(\lambda - p) + \int_{-\infty}^{\infty} d\lambda' K(\lambda - \lambda')\vartheta(\lambda')F(\lambda'|p), \tag{13}$$

where

$$\theta(\lambda) = 2\arctan(\lambda/c), \tag{14}$$

represents the scattering phase shift, related to $K$ in Eq. (10) by $K(\lambda) = \theta'(\lambda)$.

Analogously to a particle excitation, the occupation number of a certain mode with rapidity $h$ can be decreased by one, resulting into the creation of a hole excitation, with momentum and energy opposite to those of the corresponding particle excitation. Accordingly, a particle-hole excitation is characterized by a momentum $k(p,h)$ and an energy $\omega(p,h)$, given by

$$k(p,h) = k(p) - k(h), \quad \text{and} \quad \omega(p,h) = \omega(p) - \omega(h), \tag{15}$$

respectively.

## 2.3 One particle-hole kinematics at small momentum

In the small momentum limit $k \ll k_F = \pi n$, i.e., when the particle and hole are close to each other $p \simeq h$, the momentum and energy in Eq. (15) (see also Eqs. (11) and (12)) can be expressed in terms of the single particle and hole rapidities $(p,h)$ via [41]

$$k(p,h) = 2\pi(p-h)\rho_t(h), \tag{16}$$
$$\omega(p,h) = v(h)k(p,h), \tag{17}$$

where we introduced the sound velocity $v(h)$ which depends only on the rapidity of the hole excitation via

$$v(h) = \frac{\partial \omega(h)}{\partial k(h)} = \frac{h + \int_{-\infty}^{+\infty} d\lambda \, \lambda L(h,\lambda)}{\pi \rho_t(h)}, \tag{18}$$

where $L(\lambda,\mu) = -\vartheta(\mu)\partial_\mu F(\lambda|\mu)$ is related to the derivative of the back-flow function and satisfies the integral equation [41]

$$2\pi L(\lambda,\lambda'') = K(\lambda - \lambda'')\vartheta(\lambda'') + \int_{-\infty}^{+\infty} d\lambda' \, K(\lambda - \lambda')\vartheta(\lambda')L(\lambda',\lambda''). \tag{19}$$

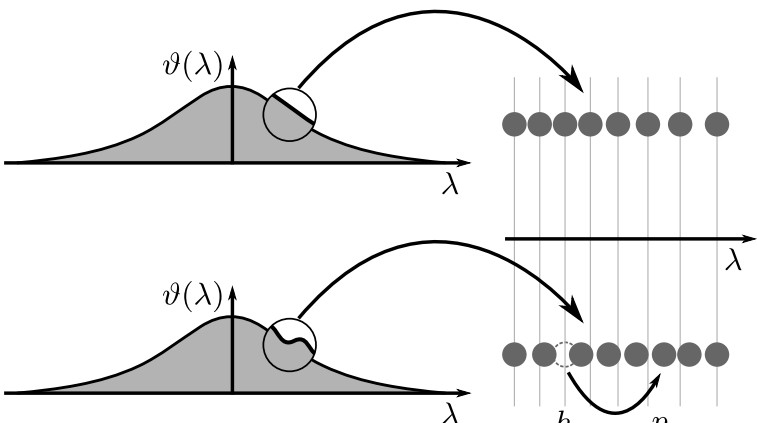

Figure 2: The filling function $\vartheta(\lambda)$, depicted on the upper part of the figure as a function of $\lambda$, describes how densely modes with different rapidities (indicated by the sequence of circles and thin vertical lines on the right, along the rapidity axis) are populated. The filling function of an excited state depicted in the lower part of the figure is macroscopically the same as the one above. However, by zooming in the region of the particle-hole excitation, indicated by the arrow on the right, reveals both a displacement of one quasi-particle from the position it was originally occupying (dashed circle) and a small shift in the (allowed) quasi-momenta of all the other particles, as indicated by the filled circles on the right. This shift compared to the original allowed rapidities (vertical lines) is described by the back-flow function $F(\lambda|p,h)$ and it contributes to the momentum and energy of the excited state as shown in Eqs. (11), (12) and (15).

## 2.4 The fluctuation-dissipation ratio

In order to introduce the fluctuation-dissipation ratio mentioned in the Introduction, we define the symmetrized correlation function $C_+$, which corresponds to the connected expectation value of the anticommutator of the particle densities at different space-time points on a generic stationary state, i.e.,

$$
\begin{aligned}
C_+(x,t) &= \left\langle \frac{\rho(x,t)\rho(0,0)+\rho(0,0)\rho(x,t)}{2} \right\rangle - \langle\rho(0,0)\rangle^2 \\
&= \frac{C(x,t)+C(-x,-t)}{2},
\end{aligned}
\tag{20}
$$

where translational invariance in both space and time of a homogeneous stationary state has been used. Although invariance under time translation is broken by a quench, one expects to recover it in the long time stationary state. The variation of the average density $\langle\rho(x,t)\rangle$ in response to a local change at, say, point $x'$ and time $t'$ in the chemical potential $\mu$ (which is conjugate to the density $\rho$) is quantified by the response function $R(x-x', t-t')$, which, via the Kubo relation [34], can be expressed as the expectation value of the commutator

$$
R(x,t) = iu(t)\langle[\rho(x,t),\rho(0,0)]\rangle,
\tag{21}
$$

where $u(t<0)=0$ and $u(t>0)=1$ is the Heaviside function. Using the following definition of the Fourier transform $f(\omega,t)$ of a function $f(x,t)$,

$$
f(k,\omega) = \int dx \int_{-\infty}^{+\infty} dt\, e^{i(kx-\omega t)} f(x,t),
\tag{22}
$$

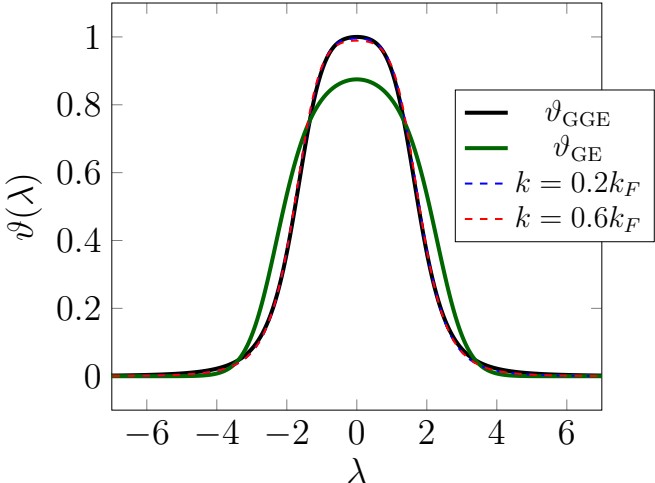

Figure 3: Comparison between the distributions $\vartheta_{\mathrm{num}}(\lambda)$ (dashed lines) inferred via Eq. (33) on the basis of the knowledge of the dynamic structure factor $S(k, \omega)$ (with $k = 0.2\,k_F$ and $0.6\,k_F$, see the left panel in Fig. 6) and the exact $\vartheta_{\mathrm{GGE}}(\lambda)$ for an interaction quench in the Lieb-Liniger model, given in Eqs. (40) and (41) at $c = 2$ and $n = 1$ (black solid line). The data at $k = 0.2\,k_F$ are indistinguishable from the exact analytical distribution on this scale. The thermal distribution $\vartheta_{\mathrm{GE}}(\lambda)$ with the average energy density and particle density appropriate for the quench under consideration is also reported for comparison (green solid line).

one can easily show that

$$\mathrm{Im}\,R(k, \omega) = \frac{S(k, \omega) - S(k, -\omega)}{2}, \tag{23}$$

where $S(k, \omega)$ is the DSF, i.e., the Fourier transform of the density-density correlations

$$C(x, t) = C(-x, t) = \langle \rho(x, t)\rho(0, 0)\rangle - \langle \rho(0, 0)\rangle^2 \tag{24}$$

in the stationary state.

In the grand canonical equilibrium with density matrix $\rho_{\mathrm{GE}} = \mathrm{e}^{-\beta(H - \mu N)}$, the cyclic property of the trace implies the fluctuation-dissipation theorem [34], i.e.,

$$S(k, -\omega) = \mathrm{e}^{-\beta\omega} S(k, \omega); \tag{25}$$

this equality establishes a sort of universal relationship between the correlation and response functions introduced above, better expressed by the fact that their fluctuation-dissipation ratio (FDR) [36, 57] satisfies

$$\Xi(k, \omega)_T \equiv \frac{\mathrm{Im}\,R(k, \omega)}{C_+(k, \omega)} = \frac{S(k, \omega) - S(k, -\omega)}{S(k, \omega) + S(k, -\omega)} = \tanh(\beta\omega/2), \tag{26}$$

as a consequence of Eq. (25). Accordingly, by determining the FDR $\Xi(k, \omega)$, one can infer the temperature $\beta^{-1}$ of the system only on the basis of dynamical quantities which can in principle be measured in an experiment.

## 2.5 Determining the effective temperatures from a generalized fluctuation-dissipation ratio

In Ref. [38] it was argued that for those integrable systems which can be mapped onto non-interacting models, the FDR in Eq. (26) can actually be used to extract the GGE effective

temperatures $\beta(\lambda)^{-1}$. Here we show that this observation generalizes to interacting integrable models. Our approach reads as follows: given an initial condition, we measure the correlation and response functions long after the quench and, either from their ratio $\Xi$ or directly from the DSF, we infer the distribution $\vartheta_{\text{GGE}}$. We stress that despite the presence of the integration over $x$ in Eq. (3), the observable $S(k, \omega)$ is local for all finite values of $k$ and for all initial states $|\Psi_0\rangle$ that respect the cluster decomposition property [58, 59]. Therefore its long time limit is given by the GGE ensemble.

In order to implement our protocol we need a relationship between $\Xi(k, \omega)$ and $\vartheta_{\text{GGE}}(\lambda)$ which we parametrize as

$$\vartheta_{\text{GGE}}(\lambda) = \frac{1}{1 + e^{\beta(\lambda)(\omega(\lambda) - \bar{\mu})}}, \tag{27}$$

where the effective chemical potential $\bar{\mu}$ is defined by

$$\bar{\mu} = \omega(\bar{\lambda}_F), \tag{28}$$

and $\bar{\lambda}_F$ generalizes the equilibrium Fermi mode out of equilibrium, with $\vartheta_{\text{GGE}}(\bar{\lambda}_F) = 1/2$. In our setting we work with the fixed density $n = N/L = 1$ and therefore both $\lambda_F$ and $\bar{\mu}$ are found from this condition by solving a pair of coupled integral equations (7) and (9).

Note that due to the interactions among the excitations, the mode energy $\omega(\lambda)$ in Eq. (27), defined in Eq. (12), is also a non-trivial functional of $\vartheta(\lambda)$ and therefore of $\beta(\lambda)$. It was established in Ref. [41] that at small momentum $k \ll k_F$ a generalized formulation of the fluctuation-dissipation theorem (26) holds. For the present purpose an additional reformulation of this relation is needed and in Appendix A we prove that it reads

$$\Xi(k, \omega) = \tanh\left(\frac{k}{2} \frac{\partial\left[\beta(\lambda)(\omega(\lambda) - \bar{\mu})\right]}{\partial k(\lambda)}\right)\Bigg|_{\lambda = \lambda(k, \omega)} + \mathcal{O}(k^2), \tag{29}$$

with the momentum $k(\lambda)$ given in equation (11). This equation follows from the fact that a perturbation with small momentum $k \ll k_F$ but arbitrary energy $\omega$ can only create a single particle-hole excitation, while multiple particle-hole excitations require probing fluctuations and perturbations with larger momenta. The value $\lambda(k, \omega)$ of the rapidity at which the r.h.s. of Eq. (29) is evaluated is fixed by the relationship (17) between the energy and momentum of the relevant single particle-hole excitation and it is therefore such that

$$\omega = v(\lambda(k, \omega))k, \tag{30}$$

with the sound velocity $v = \frac{\partial \omega}{\partial k}$ given by Eq. (18). Note that in the thermal case $\beta(\lambda) = \beta$, formula (29) reduces to the known result. Indeed using Eq. (9) and the fact that $\omega = k\frac{\partial \omega}{\partial k}$ at low momentum, we obtain

$$\Xi(k, \omega)_T = \tanh\left(\beta \frac{k}{2} \frac{\partial \omega(\lambda)}{\partial k(\lambda)}\right)\Bigg|_{\lambda = \lambda(k, \omega)} = \tanh\left(\beta \omega/2\right). \tag{31}$$

In order to determine the inverse temperatures $\beta(\lambda)$ and therefore the mode distribution $\vartheta(\lambda) = [1 + e^{\beta(\lambda)(\omega(\lambda) - \bar{\mu})}]^{-1}$ we write Eq. (29) as

$$\Xi(k, \omega) = \tanh\left(\frac{k}{2} \frac{\partial_\lambda\left[\beta(\lambda)(\omega(\lambda) - \bar{\mu})\right]}{2\pi\rho_t(\lambda)}\right)\Bigg|_{\lambda = \lambda(k, \omega)} + \mathcal{O}(k^2). \tag{32}$$

Solving this formula for $\beta(\lambda)$ and neglecting all the corrections due to finite $k$ by taking the

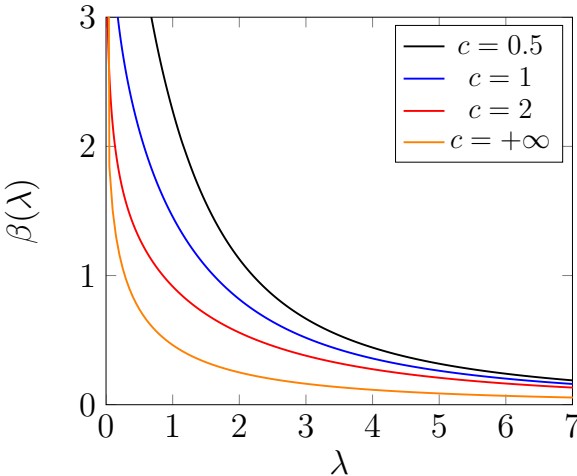

Figure 4: Dependence of the inverse effective temperatures $\beta(\lambda)$ on the rapidity $\lambda$ at long times after four quenches from the BEC state to the interacting theory with coupling $c = 0.5, 1, 2, +\infty$ (from top to bottom with $n = 1$), as obtained from the exact $\vartheta(\lambda)$ reported in Eq. (40). Note that $\beta(\lambda)$ diverges as $\log \lambda^2$ at small $\lambda$, signaling that small momenta are effectively distributed as in the ground state of the model, while the temperature describing large momenta is infinite due to the presence of high-energy excitations induced by the sudden quench. We stress that a purely thermal state would be characterized by $\beta(\lambda) = \beta$.

limit $k \to 0$ leads to:

$$\beta(\lambda) = \frac{1}{(\omega(\lambda) - \bar{\mu})} \lim_{k \to 0} \left( \frac{2}{k} \int^{\lambda} d\lambda' \, 2\pi \rho_t(\lambda') \operatorname{arctanh} \Xi(k, k \, v(\lambda')) + \text{const.} \right). \quad (33)$$

This expression can also be conveniently re-written as an integral over the energies $\omega$ by using Eq. (30)

$$\beta(\lambda) = \frac{1}{(\omega(\lambda) - \bar{\mu})} \lim_{k \to 0} \left( \frac{2}{k^2} \int^{k v(\lambda)} d\omega \, 2\pi \frac{\rho_t(\lambda(k, \omega))}{v'(\lambda(k, \omega))} \operatorname{arctanh} \Xi(k, \omega) + \text{const} \right), \quad (34)$$

where $v'(\lambda)$ indicates the derivative of $v(\lambda)$ and $\lambda(k, \omega)$ is given by Eq. (30).

Our protocol to determine $\beta(\lambda)$ after a quantum quench is as follows: given the DSF $S(k, \omega)$ (see Eq. (3)) at late times after the quench, the right hand side in Eq. (33) or (34) can be computed once the total density $\rho_t(\lambda)$, the sound velocity $v(\lambda)$, and the mode energy $\omega(\lambda)$ are known. These functions are given by integral equations depending on $\vartheta(\lambda) = (1 + e^{\beta(\lambda)(\omega(\lambda) - \bar{\mu})})^{-1}$ (see Eq. (9) for $\rho_t$ and Eqs. (18), (19) for $v(\lambda)$ and Eq. (12) for $\omega(\lambda)$) and therefore Eq. (33) or (34) must be computed iteratively on $\vartheta(\lambda)$. Moreover, the remaining integration constant in Eqs. (33) and (34) has to be fixed iteratively by imposing that the density of the gas calculated according to Eq. (7) matches its actual value. Note that although the equivalence in Eqs. (33) and (34) holds only in the limit $k \to 0$, the finite $k$ corrections are of $\mathcal{O}(k^2)$ as in Eq. (29), which makes it then possible to compute $\vartheta(\lambda)$ by using the DSF at small but finite values of $k$.

## 3 Application to the quench from the BEC state

In order to demonstrate the applicability of Eq. (33) (or, equivalently, Eq. (34)) for determining $\vartheta_{\text{GGE}}$ after a quantum quench on the basis of the knowledge of the DSF $S_t(k, \omega)$ (see

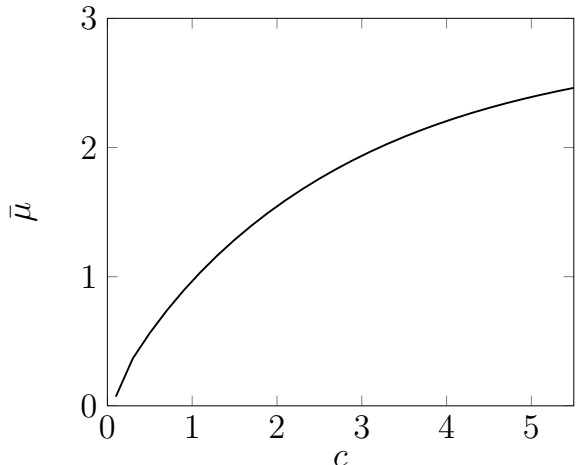

Figure 5: Dependence of the effective chemical potential $\bar{\mu}$ on the final value of the interaction $c$. At large $c$, $\bar{\mu}$ asymptotically approaches the analytically predicted value $\bar{\mu} \to 4$ (see Appendix B) while it vanishes for small $c$ (shallow quench limit) as expected for the BEC state (ground state of free bosons).

Eq. (3)) in the large time limit we "simulate" a particular quantum quench and we compute $S_t(k,\omega)$ numerically in the stationary state. In particular, we prepare the gas in the spatially homogeneous ground state $|\Psi_0\rangle$ of the Hamiltonian $H$ in Eq. (6) with $c = 0$ (the so-called BEC state) and density $n = 1$, with $N = L = 20$ and we let it evolve with the same Hamiltonian but with $c \neq 0$. For this quench, the overlaps $\langle\Psi_0|\alpha\rangle$ between the initial state $|\Psi_0\rangle$ and the eigenstates $|\alpha\rangle$ of $H$ with non-vanishing $c$ are known [60] and, accordingly, we can follow the evolution from the initial state and access the eventual GGE as time goes by. We consider the density-density correlations on the time-evolved initial state

$$C_t(x,t') = \langle\Psi_0|\rho(x,t+t')\rho(0,t)|\Psi_0\rangle - n^2, \tag{35}$$

(where we used the fact that the initial state is homogeneous at all times and therefore $\langle\Psi_0|\rho(x,t)|\Psi_0\rangle = n$) and consider the DSF $S_t(k,\omega)$ obtained from its Fourier transform with respect to the space and time-delay $t'$, with fixed $t$, according to Eq. (22) (see also Eq. (3)). Inserting twice the resolution of the identity

$$1 = \sum_{\alpha \in \text{span}(H)} |\alpha\rangle\langle\alpha|, \tag{36}$$

in terms of the (Bethe) eigenstates $\{|\alpha\rangle\}$ of the post-quench Hamiltonian $H$ with eigenvalues $\{E_\alpha\}$, the DSF can be written as

$$S_t(k,\omega) = \sum_{\alpha,\beta} \langle\alpha|\Psi_0\rangle\langle\Psi_0|\beta\rangle e^{-it(E_\alpha - E_\beta)} S_{\alpha,\beta}(k,\omega), \tag{37}$$

where $S_{\alpha,\beta}(k,\omega)$ is the DSF determined between the Hamiltonian eigenstates $|\alpha\rangle$ and $|\beta\rangle$, i.e., the Fourier transform of $\langle\beta|\rho(x,t')\rho(0,0)|\alpha\rangle - n^2$ (with $n = 1$ in this specific quench). While, in principle, we could calculate the DSF of the density fluctuations in the GGE by letting $t \to \infty$ in Eq. (37), we actually assume that one can access $S_t(k,\omega)$ only up to a moderately long time $T$ (with respect to the typical relaxation time scale of the system, see Appendix D), as it is the case both in experimental realizations and in numerical calculations. We then approximate the actual asymptotic DSF $S(k,\omega) \equiv S_{t\to\infty}(k,\omega)$ with the one obtained by averaging $S_t(k,\omega)$

over time, in order to average out possible oscillations:

$$\bar{S}_T(k,\omega) = \frac{1}{T} \int_0^T dt\, S_t(k,\omega). \tag{38}$$

In the limit of large $T$, $\bar{S}_T$ becomes identical to the density-density correlation in the GGE up to finite-size corrections and it can actually be expressed as an average over the so-called diagonal ensemble [33]

$$S(k,\omega) \equiv \lim_{T\to\infty} \bar{S}_T(k,\omega) = \sum_\alpha |\langle\alpha|\Psi_0\rangle|^2 S_{\alpha,\alpha}(k,\omega). \tag{39}$$

In Appendix E we discuss how this quantity can in principle be measured in experiments. In the next subsection, instead, we proceed to its numerical computation based on the ABACUS code [53, 61–63] (see also Appendix F) which allows us to determine the r.h.s. of Eq. (39) and extract, as explained in the previous subsection, the numerical estimate $\vartheta_{num}$ of the distribution $\vartheta$. Then, $\vartheta_{num}$ can be compared with the exact $\vartheta_{GGE}$, analitically determined in Ref. [60] for this particular quench and which can be written as

$$\vartheta_{GGE}(\lambda) = \frac{a(\lambda)}{1+a(\lambda)}, \tag{40}$$

with the function $a(\lambda)$ given by

$$a(\lambda) = \frac{2\pi/\gamma}{(\lambda/c)\sinh(2\pi\lambda/c)} \left| I_{1\pm 2i(\lambda/c)}(4/\sqrt{\gamma}) \right|^2, \tag{41}$$

where $I_\alpha(z)$ is the modified Bessel function of the first kind and $\gamma = c/n$.

The corresponding effective temperatures $\beta(\lambda)$ can now be easily calculated as $\beta(\lambda)(\omega(\lambda)-\bar{\mu}) = -\ln a(\lambda)$ (see Eq. (27)) and $\omega(\lambda)$ given by Eq. (12). Figure 4 and 5 show the inverse effective temperatures $\beta(\lambda)$ and the effective chemical potential $\bar{\mu}$ for quenches from the BEC state to different values of the couplings, parametrized by different values of $c = \gamma$ (we recall that we have choose unitary density of the gas $n = 1$).

## 3.1 Numerical results

Figure 6 shows the dependence of the DSF $S(k,\omega)$ on the frequency $\omega$ within the range $|\omega| \leq 3\,\omega_F$, where $\omega_F = k_F^2 = \pi^2 n^2$ is the Fermi energy and for three values of the wave-vector $k = 0.2\,k_F, 0.4\,k_F, 0.6\,k_F$. In particular, the curves on the left panel are calculated numerically with the ABACUS code ($N = L = 20$) after the quench to $c = 2$, while those on the right panel correspond to the analytic expressions which can be obtained in the thermodynamic limit with $n = 1$ and $c \to \infty$ as briefly discussed in Appendix B.

Based on the numerical data for $S(k,\omega)$ reported on the left panel of Fig. 6 — which mimic the result of a possible scattering experiment discussed in Appendix E — we can now use Eq. (33) in order to extract the numerical estimate $\vartheta_{num}$ of the distribution $\vartheta$ for a quench from the BEC state to $c = 2$, with density $n = 1$. The resulting $\vartheta_{num}$ is reported in Fig. 3 as an even function of the rapidity $\lambda$, for two values of the wave-vector $k$ at which data was obtained. The solid line in Fig. 3 corresponds to the exact expression of $\vartheta$ which follows from Eqs. (40) and (41). The agreement between the two numerical estimates $\vartheta_{num}$ and $\vartheta$ is remarkably good and discrepancies are barely visible on this scale. Note also that while Eq. (33) is actually valid for $k \ll k_F$, the dependence of $\vartheta_{num}$ on $k$ is practically irrelevant up to the largest value of $k$ considered here. We point out that the numerical estimate $\vartheta_{num}$ also allows one to distinguish between the GGE distribution and its thermal approximations,

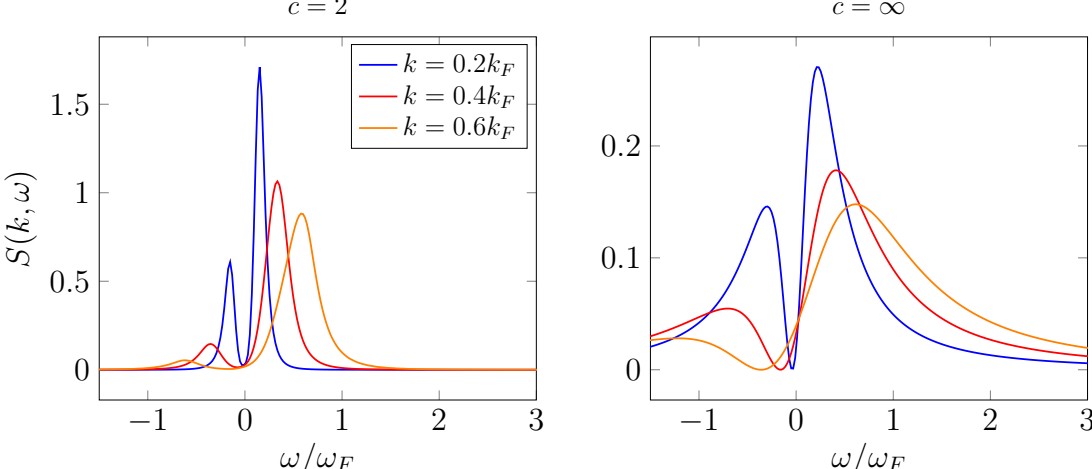

Figure 6: Dependence of the dynamic structure factor $S(k,\omega)$ of the post-quench diagonal ensemble (39) on the frequency $\omega/\omega_F$ with the Fermi energy $\omega_F = k_F^2$, for three values of the wave-vector $k = 0.2\,k_F, 0.4\,k_F$, and $0.6\,k_F$. The left and the right panels correspond to a quench to $c = 2$ and $c \to \infty$, respectively. In the left panel the curves are obtained via a numerical computation with the ABACUS code for $N = L = 20$ and $c = 2$ while in the right panel the curves correspond to the analytic prediction for the case $c \to \infty$ in the thermodynamic limit $N \to \infty$, $L \to \infty$ with $n = 1$, derived in Ref. [64].

the GE distribution $\vartheta_{\mathrm{GE}}$ at fixed density $n = 1$ where only the Hamiltonian $H$ is used as local conserved charge after the quench. Note that the deviations of the estimate $\vartheta_{\mathrm{num}}$ based on the numerical data from the exact result are due to finite-size effects and the systematic error introduced by using a small but finite $k$ (see also Fig. 10).

In order to determine $\vartheta_{\mathrm{num}}$ from the fluctuation-dissipation ratio $\Xi(k,\omega)$ we use Eq. (33) (or equivalently Eq. (34)), and the relation between $\Xi(k,\omega)$ and $S(k,\omega)$ given in Eq. (26). Figure 7 shows the dependence of $\Xi(k,\omega)$ on $\omega$, for the same three values of $k$, quenches and data considered in Fig. 6. In particular, the left and right panels correspond to quenches to $c = 2$ and $c \to \infty$, respectively. In the latter case (right), the curves clearly show that $\Xi(k,\omega)$ vanishes, for fixed $k$, as $|\omega| \to \infty$, as it was analytically shown in Ref. [64] and the same behavior is expected for any quench to $c > 0$ (left) despite the fact that numerical artifacts (due to the truncated diagonal ensemble used in order to numerically compute the dynamical structure factor as in (39)) hide it.

# 4 Conclusions and perspectives

In this work we have presented a method which allows one to completely determine the stationary state of a quantum integrable model from the knowledge of a dynamic structure factor. This approach is solely based on experimentally accessible observables, it characterizes the stationary state via the mode occupation numbers, and it does not rely on the knowledge of the conserved charges or their expectation values. For the interaction quench considered here, it circumvents the technical difficulties encountered in the direct construction of the GGE ensemble based on these charges.

In most of the experiments with ultracold atoms, the possible thermal nature of their stationary state is typically probed by measuring the momentum distribution of the atoms. However, more detailed information can be inferred from spatially or temporally resolved density

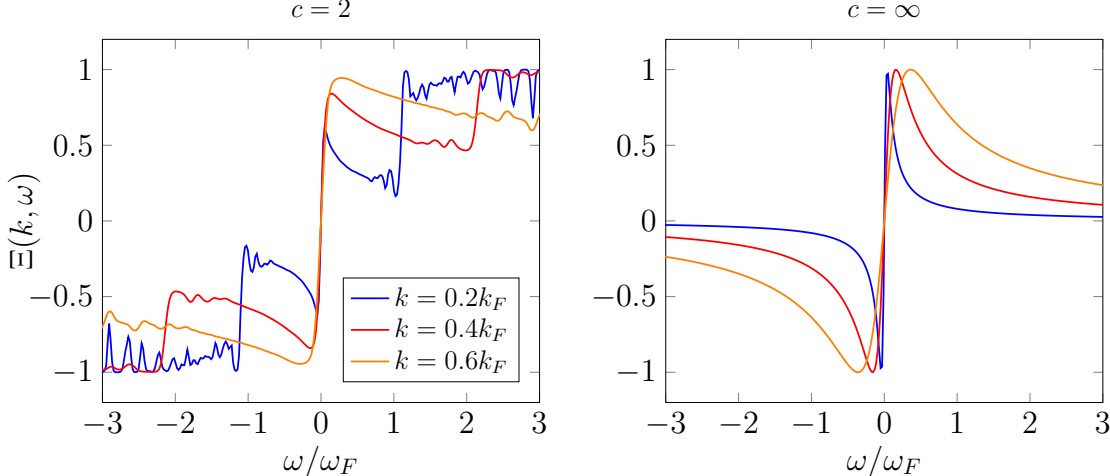

Figure 7: Dependence of the fluctuation-dissipation ratio $\Xi(k,\omega)$ on $\omega/\omega_F$ with the Fermi energy $\omega_F = k_F^2$ as calculated from the post-quench diagonal ensemble (39) for the three different values of $k$ indicated in the legend, corresponding to the same data and quenches as in Fig. 6. The left and right panels correspond to a quench to $c = 2$ and $c \to \infty$, respectively. The abrupt increase in $\Xi(k,\omega)$ at $c = 2$ (for example at $k = 0.2k_F$ around $\omega/\omega_F = \pm 1$) and the rapid oscillations are a numerical artifact due to the numerical truncation of the sum over the diagonal ensemble (39).

correlation functions, such as those that we propose to study here. For example, the phase correlation between two halves of the gas after they have been split in space was used in Ref. [9] in order to monitor the possible thermalization of the system, while currently available experimental techniques allow the determination of structure factors via Bragg spectroscopy (see, e.g., Refs. [43, 44]). Although it has not yet been measured experimentally, we expect the post-quench time evolution of the dynamic structure factor $S(k,\omega)$ of density correlations discussed here to be within experimental reach. Our protocol allows one to extract the full steady state from simple measurements on the system: accordingly, cold atoms experiments or numerical algorithms might permit access to quantities that have been out of reach. The approach discussed here also provides a powerful method to quantify and characterize non-thermal fluctuations in a generic cold atomic gas.

An interesting open question is how to extend the analysis presented here to systems with different particle types, like the attractive Lieb-Liniger model, where different bound states of particles can coexist in the same eigenstate, or lattice models like the XXZ spin chain. Moreover, it will be interesting to test our approach in non-homogenous non-equilibrium conditions, like the ones recently treated, e.g. in Refs. [51, 52]. Another area in which a similar analysis could apply are integrable field theories [14, 21, 65, 66]. We leave these questions open for future studies.

**Acknowledgements** L. F. Cugliandolo is a member of Institut Universitaire de France. We are indebted to J.-S. Caux for the use of the ABACUS software package.

**Funding information** JDN acknowledges support by LabEx ENS-ICFP:ANR-10-LABX-0010/ANR-10-IDEX-0001-02 PSL*. MP acknowledges financial support from the Polish National Science Centre (NCN) under FUGA grant 2015/16/S/ST2/00448. LF acknowledges support by the European Research Council under the European Union's 7th Framework Programme (FP/2007-2013/ERC Grant Agreement 307087-SPARCS). R. Konik was supported by the U.S. Depart-

ment of Energy, Office of Basic Energy Sciences, under Contract No. DE- AC02-98CH10886.

## A  Proof of an equivalence

In this Appendix we show that Eq. (29) follows from Eq. (102) of Ref. [41], which we refer to for the notation and the definition of the various quantities involved in the proof presented below. In particular, in Ref. [41] it was shown that with the notations introduced in Eq. (26),

$$\Xi(k,\omega) = \tanh(\mathscr{F}(k,\omega)/2) + \mathcal{O}(k^2), \tag{42}$$

with $\mathscr{F}(k,\omega)$ defined as (see Eq. (102) of Ref. [41])

$$\mathscr{F}(k,\omega) = \frac{k\,\partial_\lambda \epsilon_d(\lambda)}{2\pi\rho_t(\lambda)}, \tag{43}$$

where $\lambda = \lambda(\omega,k)$ is given by Eq. (30) here, while $\epsilon_d$ is obtained by dressing the bare GGE driving term $\epsilon_0(\lambda)$

$$\epsilon_d(\lambda) = \epsilon_0(\lambda) - \int_{-\infty}^{\infty} d\lambda'\, \partial_\lambda \epsilon_0(\lambda') F(\lambda'|\lambda)\vartheta(\lambda'), \tag{44}$$

with

$$\epsilon_0(\lambda) = \sum_n \mu_n \epsilon_0^{(n)}(\lambda), \tag{45}$$

where $\epsilon_0^{(n)}(\lambda)$ is the eigenvalue of the charge $Q_n$ on a Bethe state with a single mode $N = 1$ (also referred to as bare charge, see Ref. [41] for its definition). In order to show that Eq. (29) holds as a consequence of Eq. (42), we need to prove that

$$\mathscr{F}(k,\omega) = \frac{k\,\partial_\lambda \epsilon(\lambda)}{2\pi\rho_t(\lambda)}\bigg|_{\lambda=\lambda(k,\omega)} \quad \text{with} \quad \epsilon(\lambda) = \beta(\lambda)[\omega(\lambda) - \bar{\mu}], \tag{46}$$

i.e., we shall show that $\partial_\lambda \epsilon_d(\lambda) = \partial_\lambda \epsilon(\lambda)$. To do so one needs to prove that the two dressing relations, given a bare energy $\epsilon_0(\lambda)$ and its dressed counterpart $\epsilon(\lambda)$,

$$\epsilon(\lambda) = \epsilon_0(\lambda) + \int_{-\infty}^{\infty} \frac{d\lambda'}{2\pi} K(\lambda - \lambda') \log(1 + e^{\epsilon(\lambda')}), \tag{47}$$

and the one given by

$$\epsilon_d(\lambda) = \epsilon_0(\lambda) - \int_{-\infty}^{\infty} d\lambda'\, \partial_\lambda \epsilon_0(\lambda') F(\lambda'|\lambda)\vartheta(\lambda'), \tag{48}$$

are equivalent up to a constant shift $\epsilon_d = \epsilon + \text{const}$.

Taking the derivative of Eq. (47) with respect to $\lambda$ we obtain

$$\partial_\lambda \epsilon(\lambda) = \partial_\lambda \epsilon_0(\lambda) + \int_{-\infty}^{\infty} \frac{d\lambda'}{2\pi} K(\lambda - \lambda'')\vartheta(\lambda')\partial_\lambda \epsilon(\lambda'). \tag{49}$$

On the other hand by doing the same to Eq. (48) we find

$$\partial_\lambda \epsilon_d(\lambda) = \partial_\lambda \epsilon_0(\lambda) - \int_{-\infty}^{+\infty} d\lambda'\, \partial_\lambda \epsilon_0(\lambda')\partial_\lambda F(\lambda'|\lambda)\vartheta(\lambda'). \tag{50}$$

As shown in Ref. [41], one has

$$\partial_\lambda F(\lambda'|\lambda) = -L(\lambda',\lambda)/\vartheta(\lambda) = -L(\lambda,\lambda')/\vartheta(\lambda'), \tag{51}$$

which can be used in order to write Eq. (50) as

$$\partial_\lambda \epsilon_d(\lambda) = \partial_\lambda \epsilon_0(\lambda) + \int_{-\infty}^{\infty} d\lambda' \, L(\lambda,\lambda') \partial_\lambda \epsilon_0(\lambda') \tag{52}$$

by using that $(\delta + L) = (\delta - \frac{K}{2\pi}\vartheta)^{-1}$ (where $\delta$ is the Dirac delta operator and this relation has to be understood in terms of generalized distribution functions) one then finds

$$\int_{-\infty}^{+\infty} d\lambda' \left[ \delta(\lambda - \lambda') - \frac{K(\lambda - \lambda')}{2\pi}\vartheta(\lambda') \right] \partial_\lambda \epsilon_d(\lambda') = \partial_\lambda \epsilon_0(\lambda), \tag{53}$$

which is actually the same as Eq. (49) and therefore $\partial_\lambda \epsilon_d(\lambda) = \partial_\lambda \epsilon(\lambda)$.

## B  The free limit $c \to \infty$

For $c \to \infty$, one has $v(\lambda) = 2\lambda$ and $2\pi\rho_t(\lambda) = 1$. Accordingly, from Eqs. (33) and (27) we get

$$\beta(\lambda)[\lambda^2 - \bar{\mu}] = \lim_{k \to 0} \left[ \frac{2}{k} \int^\lambda d\lambda' \, \text{arctanh} \, \Xi(k, k\,v(\lambda')) + \text{const} \right]. \tag{54}$$

The DSF $S(k,\omega)$ is known exactly in this case from Ref. [64] (here we set the density $n = 1$) and turns out to be:

$$S(k,\omega) = \frac{8|k|(k^2 + \omega)^2}{[(4k)^2 + (k^2 - \omega)^2][(4k)^2 + (k^2 + \omega)^2]}, \tag{55}$$

and therefore

$$\Xi(k,\omega) = \frac{S(k,\omega) - S(k,-\omega)}{S(k,\omega) + S(k,-\omega)} = \frac{2k^2\omega}{k^4 + \omega^2}. \tag{56}$$

Note the decay $\propto \omega^{-1}$ of $\Xi(k,\omega)$ at large $\omega$, while one expects it to be $\propto \omega^{-2}$ in the interacting case. Substituting this expression for $\Xi(k,\omega)$ into Eq. (54), we obtain

$$\beta(\lambda)(\lambda^2 - \bar{\mu}) = 2 + \ln(4\lambda^2) + \text{const.}, \tag{57}$$

where the integral was calculated with the principal value. Accordingly, by choosing

$$\text{const.} = -2 - 2\ln 4, \tag{58}$$

we obtain the exact GGE post-quench distribution of momenta

$$\vartheta(\lambda) = \frac{1}{1 + e^{\beta(\lambda)(\lambda^2 - \bar{\mu})}} = \frac{1}{1 + (\lambda/2)^2} \tag{59}$$

found in Ref. [64]. This result shows that in the case $c = \infty$, one has $\beta(\lambda) = \left[\ln(\lambda/2)^2\right]/(\lambda^2 - \bar{\mu})$ and $\bar{\mu} = 4$.

## C  The limit of large but finite $c$

In Appendix B we considered the limit $c \to \infty$. It is possible to develop this further and demonstrate how to extract $\vartheta(\lambda)$ from $S(k, \omega)$ up to $1/c^2$ corrections. We will show that this is the case even if the momentum $k$ of the dynamic structure factor is finite. This is possible because at large $c$ the two (and greater) particle-hole contributions to $S(k, \omega)$ are suppressed by $1/c^2$ regardless of the magnitude of $k$. This fact was used by two of the authors here in Ref. [41] to develop an expression for $S(k, \omega)$ valid for arbitrary $\vartheta(\lambda)$ and arbitrary $(k, \omega)$ up to $1/c^2$ corrections (see Eq. (5.8) of Ref. [67]), which turns out to be

$$S(k, \omega) = \frac{2}{\pi} \left[ \pi \frac{1 + 6n/c}{4|k|} + \frac{1}{2c} \frac{k}{|k|} \mathrm{PV} \int_{-\infty}^{\infty} d\lambda \frac{\vartheta(\lambda + p) - \vartheta(\lambda + h)}{\lambda} \right] \vartheta(h)[1 - \vartheta(p)]$$
$$+ \mathcal{O}(1/c^2), \quad (60)$$

with

$$p = \frac{k}{2(1 + 2n/c)} + \frac{\omega(1 + 2n/c)}{2k}, \quad (61)$$

$$h = -\frac{k}{2(1 + 2n/c)} + \frac{\omega(1 + 2n/c)}{2k}. \quad (62)$$

We see that $S(k, \omega)$ involves $\vartheta(\lambda)$ in a straightforward fashion. Even though $\vartheta(\lambda)$ appears in an integral in the above expression, we can, by using the $1/c$ expansion of $\vartheta$,

$$\vartheta(\lambda) = \vartheta_0(\lambda) + \frac{1}{c}\vartheta_1(\lambda) + \mathcal{O}(1/c^2), \quad (63)$$

arrive at a simple expression for it in terms of $S(k, \omega)$. As part of this we will need to know how the $1/c$ corrections to $\omega(\lambda)$ and $k(\lambda)$ (Eqs. (12) and (11)) appear. These, however, are also straightforward to develop because the scattering kernel $K(\lambda)$ becomes particularly simple at leading order in $1/c$:

$$K(\lambda) = \frac{2}{c} + \mathcal{O}(1/c^2). \quad (64)$$

To proceed we first solve Eq. (60) for $\vartheta(\lambda)$ at leading order in $1/c$. To simplify this we consider the elastic limit, where $\omega = 0$. In this case $p = -h$ and, using Eq. (64), it is straightforward to show that $k$ and $p$ are related by:

$$k = 2p(1 + 2n/c) + \mathcal{O}(1/c^2). \quad (65)$$

In this particular case we then obtain

$$\vartheta(p) = \frac{1}{2}\left[ 1 + \sqrt{1 - 16|p|S(2p, 0)} \right] + \mathcal{O}(1/c). \quad (66)$$

This expression allows us to rewrite Eq. (60) as follows

$$S(k, 0) = \frac{1}{F(k)}\vartheta(p)(1 - \vartheta(p));$$
$$F^{-1}(k) = \frac{2}{\pi}\left\{ \pi \frac{1 + \frac{6n}{c}}{4|k|} + \frac{n \, \mathrm{sgn}(k)}{2c} \mathrm{PV} \int_{-\infty}^{\infty} \frac{d\lambda}{\lambda} \right.$$
$$\left. \left[ \left(1 - 16\left|\lambda + \frac{k}{2}\right|S(\lambda + \frac{k}{2}, 0)\right)^{1/2} - \left(1 - 16\left|\lambda - \frac{k}{2}\right|S(\lambda - \frac{k}{2}, 0)\right)^{1/2} \right] \right\}. \quad (67)$$

We can then solve this to obtain $\vartheta(\lambda)$ up to $\mathcal{O}(1/c^2)$ corrections:

$$\vartheta(p) = \frac{1}{2}\left[1 + \sqrt{1 - 4F(k)S(k,0)}\right]\Bigg|_{k=2p(1+2n/c)} + \mathcal{O}(1/c^2). \tag{68}$$

We also note that in this large-$c$ limit one can recover the same equation derived for the structure factor in Ref. [38]. In fact from Eq. (61) one can deduce the following properties:

$$h_{-k,\omega} = -h_{k,\omega}, \quad p_{-k,\omega} = -p_{k,\omega}, \quad \text{and} \quad h_{-k,-\omega} = p_{k,\omega}, \quad p_{-k,-\omega} = h_{k,\omega}, \tag{69}$$

where the subscript indicates that they are the particle and hole pair determining $S(k,\omega)$. Taking into account these properties, from Eq. (60) one finds

$$S(k,\omega) = S(-k,\omega),$$

$$\frac{S(k,\omega)}{S(-k,-\omega)} = \frac{\vartheta(h_{k,\omega})\left[1 - \vartheta(p_{k,\omega})\right]}{\vartheta(p_{k,\omega})\left[1 - \vartheta(h_{k,\omega})\right]} = e^{\beta(p_{k,\omega})[\omega(p_{k,\omega}) - \bar{\mu}] - \beta(h_{k,\omega})[\omega(h_{k,\omega}) - \bar{\mu}]}, \tag{70}$$

which gives the FDT ratio of Eq. (26) in Ref. [38]. From Eq. (40) we write:

$$\vartheta(\lambda) = \frac{1}{1 + a^{-1}(\lambda)} \tag{71}$$

and an expansion at large $c$ gives:

$$a^{-1}(\lambda) = \frac{\lambda^2}{4n^2}\left(1 + 4\frac{n}{c}\right) + o\left(\frac{n}{c}\right). \tag{72}$$

Using this expression we define:

$$\bar{\lambda}_F = 2n\sqrt{\frac{c}{4n+c}}, \qquad \bar{\mu} = \frac{4n^2 c}{4n+c}, \quad \text{and} \quad \beta(\lambda) = \frac{\log a^{-1}(\lambda)}{\lambda^2 - \bar{\mu}}. \tag{73}$$

Moreover, choosing,

$$k_\lambda = \left(1 + \frac{2n}{c}\right)\left(\lambda - \frac{\bar{\mu}}{\lambda}\right) \quad \text{and} \quad \omega_\lambda = \lambda^2 - \left(\frac{\bar{\mu}}{\lambda}\right)^2 \tag{74}$$

implies

$$\frac{S(k_\lambda, \omega_\lambda)}{S(-k_\lambda, -\omega_\lambda)} = e^{2\beta(\lambda)[\omega(\lambda) - \bar{\mu}]}, \tag{75}$$

which is a condition that directly relates the measurement of $S(k,\omega)$ to $\beta(\lambda)$ via the FDT (or detailed balance) ratio, in an analogous manner to what has been done in [38] for the case $c = \infty$.

# D    Effects of a harmonic trap

In the previous sections we considered the case of a Bose gas in one spatial dimension and of infinite extent. However, in actual experiments, the cold atoms which constitute the gas are spatially confined by the presence of an optical trap which generates a one-body potential acting on the atoms and therefore an additional contribution $\sum_{i=1}^{N} V(x_i)$ in Eq. (6). This confinement typically occurs on a length scale $\ell$ which is large on the microscopic scale and eventually affects, e.g., the DSF $S(k,\omega)$ at small wave vectors $k \lesssim \ell^{-1}$. As Eqs. (33) and (34), which allow us to determine $\vartheta$ from $S(k,\omega)$ are actually based on its expansion at small

momenta, see Eq. (29), the presence of a trap may affect significantly the whole procedure and therefore it is worth considering in more detail its effect. In particular, the trap breaks the integrability of the model, as discussed further below.

A first heuristic estimate of the consequences of a trap can be obtained by considering the static structure factor

$$S(k) \equiv \int_{-\infty}^{\infty} \frac{d\omega}{2\pi} S(k, \omega), \tag{76}$$

in the ground state of the gas, as opposed to its dynamical counterpart $S(k, \omega)$. The effects on measuring $S(k)$ after release from a trap was computed in Ref. [67], where it was argued that a harmonic trap $V(x) = m\omega_{\text{trap}}^2 x^2/2$ (where $m$ is the mass of the gas particles) of characteristic frequency $\omega_{\text{trap}}$ changes the structure factor $S$ by an amount $\delta S$ which, at small momenta reads [67]

$$\delta S(k \ll k_F) = \frac{k_F}{16\pi} \left(\frac{\omega_{\text{trap}}}{\omega_F}\right)^4 \left(\frac{k_F}{k}\right)^5, \tag{77}$$

with the Fermi energy given by $\omega_F = k_F^2$. The dependence of $\delta S$ on $k^{-5}$ seemingly suggests that the actual structure factor in the presence of the trap is significantly affected at small wave vectors compared to the one in its absence. However, under actual reported experimental conditions [43, 44], the corrections implied by Eq. (77) turns out to be small even for $k_{\text{trap}} = 2\pi/L_{\text{trap}} \sim \omega_F/(\pi\omega)$ where $L_{\text{trap}}$ is the length of the trap. Reference [43] reported $\omega_{\text{trap}}/\omega_F \sim 10^{-2}$, which implies $\delta S(k_{\text{trap}}) \sim 10^{-4} k_F$.

A second characterization of the trap can be made by using the local density approximation (LDA). Such an approximation is indeed justified as the typical macrostate describing the gas after the quench is a finite energy state with short-range correlations and with a finite correlation length. The LDA as applied to the determination of the DSF, tells us that the measured $S(k, \omega)$ in a trap should be averaged over the spatial extent of the trap, i.e.,

$$S_{\text{measured}}(k, \omega) = \int_{-L_{\text{trap}}/2}^{L_{\text{trap}}/2} dx\, S(k, \omega, n(x)), \tag{78}$$

where $n(x)$ is the density of the gas at position $x$ of the trap. The density at a given point is given by

$$n(x) = \frac{1}{\sqrt{\pi}} \left(\pi n_0^2 - \frac{\omega_{\text{trap}}^2 x^2}{4}\right)^{1/2}, \tag{79}$$

where $n_0 = n(0)$ is the density at the center of the trap and $\omega_{\text{trap}}$ is the strength of the trap. This allows us to recast $S_{\text{measured}}$ in a simple form,

$$S_{\text{measured}}(k, \omega) = \int_0^1 dy \frac{y}{\sqrt{1-y^2}} S(k, \omega, n_0 y). \tag{80}$$

Note that $\omega_{\text{trap}}$ does not directly appear in this expression.

Using Eq. (80), we can see how the trap distorts the DSF $S(k, \omega)$. As an example, we explicitly consider the case $c = \infty$, for which Ref. [64] provides the exact expression of the DSF as a function of the density:

$$S(k, \omega, n) = \frac{8n^2(k^2 + \omega)|k|}{[(4nk)^2 + (k^2 - \omega)^2][(4nk)^2 + (k^2 + \omega)^2]}. \tag{81}$$

Using this expression, we plot in Fig. 8 both $S(k, \omega, n_0)$ and $S_{\text{measured}}(k, \omega)$. We see that while there are distortions induced by the presence of the trap on $S(k, \omega)$, they are relatively small.

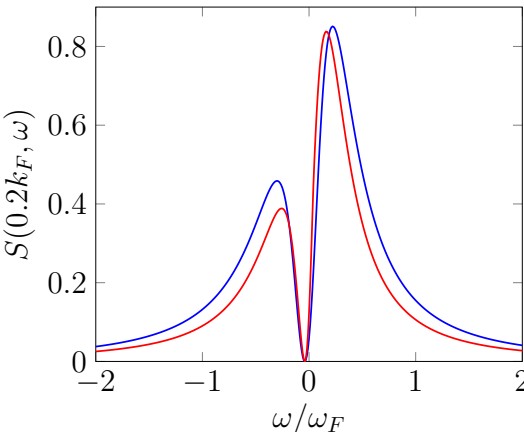

Figure 8: Plot of $S(k, \omega)$ at density $n = k_F/\pi$ in a homogeneous system (blue line) against $S(k, \omega)$ in a trap as computed using the LDA (red line). We see that while the trap effects small quantitative changes on $S(k, \omega)$, it leaves the overall line shape intact.

Unlike the zero-temperature case [68, 69], the change in lineshape of the DSF is quantitative, not qualitative.

Beyond the issue of the extent that the trap distorts the dynamic structure function, its presence gives rise to the need to address how exactly the late time dynamics of the gas is still described by a GGE, the premise of this paper. In part this is a question of time scales and strength of the integrability breaking. The integrability breaking due to the trap happens on a certain time scale, $\tau_{\text{break}} \sim \frac{1}{\omega_{\text{trap}}}$. For shallow traps $\omega_{\text{trap}} \ll \omega_F$ this time scale is much longer than the time scale, $\tau_{\text{rel}} \sim \frac{1}{\omega_F}$, governing relaxation in the system, i.e.. $\tau_{\text{break}} \gg \tau_{rel}$, we can expect the system to relax to what is known to an ensemble governed by a deformed GGE (see Refs. [30, 70, 71]). A deformed GGE is an ensemble that still possesses an infinite number of charges, but whose charges differ (parametrically in the strength of the integrability breaking) from the original. But this means that we can conclude provided integrability breaking is sufficiently weak that we expect that a time range emerges within which $S(k, \omega)$ is well described by the GGE of the original, unperturbed, model. This plateau is known typically as a prethermalization plateau [31, 72, 73].

The discussion in the previous paragraph assumed that the system was in the thermodynamic limit. However we can ask the same question on the fate of the GGE in a *finite* system (as all experimental systems are), namely does weak integrability breaking due to a trap lead to relaxation to a state governed by a Gibbs ensemble (at least up to finite-size corrections)? Here the answer is that even at infinite time, weak integrability breaking may not lead to Gibbs-like thermalization. As shown in [28], a remnant of the conserved charges survive integrability breaking (in the same spirit that integrability breaking in a classical integrable system does not destroy all invariant tori). These remnants will prevent the system from ever thermalizing to a Gibbs ensemble and might be thought of as the infinitely long-lived relatives of the deformed GGE described in Refs. [70, 71].

## E    Discussion on possible experimental realizations

In this section we discuss the possible experimental realization of the proposed method for determining the GGE state. Currently available experimental techniques allow one to realize the one-dimensional Bose gas either in the atom chip setup [54, 74] or via optical lattices [75–

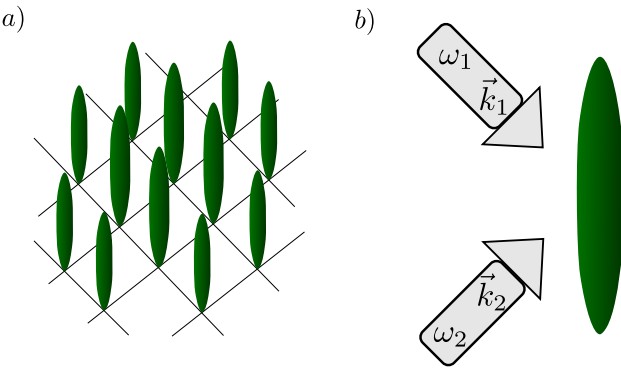

Figure 9: (a) A two-dimensional optical lattice creates an array of one-dimensional tubes. (b) The Bragg scattering perturbs the gas in each tube with energy $\omega = \omega_1 - \omega_2$ and momentum $k$ proportional to the magnitude of $|\vec{k}_1| = |\vec{k}_2|$ and depending on the relative angle between them. This figure is reproduced from Ref. [44].

77]. In the latter, the one-dimensional Bose gas is created by tightly confining the motion of the cloud of atoms along a single direction in space. The periodicity of the optical lattice results not in a single one-dimensional Bose gas, but in an two-dimensional array of "one-dimensional tubes" (see Fig. 9). A gas in each tube is effectively described by the Lieb-Liniger model but with different densities, maximal in the center of the array and decreasing outwards. This has two effects. First, the effective interaction parameter $\gamma = c/n$ varies among the array. The tubes further from the center have stronger interactions. Second, it affects the Bragg spectroscopy. Let us discuss the second effect in more detail.

In the Bragg spectroscopy experiments [43, 44, 55, 78] two slightly detuned lasers create a standing wave which perturbes the gas, c.f., Fig. 9. The momentum $k$ of the perturbation depends on the geometry, while the energy $\omega$ on the detuning. Taking the relatively small size of the array of the atoms into account we can safely assume that $k$ is constant throughout the array. Accordingly, each tube is perturbed with the same momentum and energy. However, the gas in the tubes further away from the center has a lower density and therefore it is effectively perturbed at an higher momentum relative to the density-dependent Fermi momentum $k_F = \pi n$. This has the following consequences.

Assume that the experimental conditions are chosen in such a way that the center tubes are perturbed at small momenta (with respect to their $k_F$). A measurement would then give $S(k, \omega)$ at small $k$, which is exactly what we need to recover the distribution $\vartheta(\lambda)$. However the tubes further from the center will be perturbed at higher momenta and it seems that we could not determine $\vartheta(\lambda)$ from their DSF. Fortunately, the interactions in these tubes are also stronger, their $\gamma$ is larger. Accordingly, what we obtain from the Bragg spectroscopy is a DSF at larger momenta of a stronger interacting one-dimensional Bose gas. It turns out that for a strongly interacting gas we can use a slightly different method of determining $\vartheta(\lambda)$. It is based on the perturbative expansion around $\gamma = \infty$ and in Appendix C we provide the details.

Note that these two methods for determining $\vartheta(\lambda)$ are complementary. One, the main focus of this work, relies on small-$k$ features of the DSF and is valid for all interaction strengths. The second, based on a $1/\gamma$ expansion, is applicable only for strong enough interactions ($\gamma > 10$ usually proves to be sufficient), but does not require any restriction on the momentum at which the DSF is considered. This allows us to cover the whole variety of physical situations encountered in the Bragg spectroscopy of the one-dimensional Bose gas created in optical lattices.

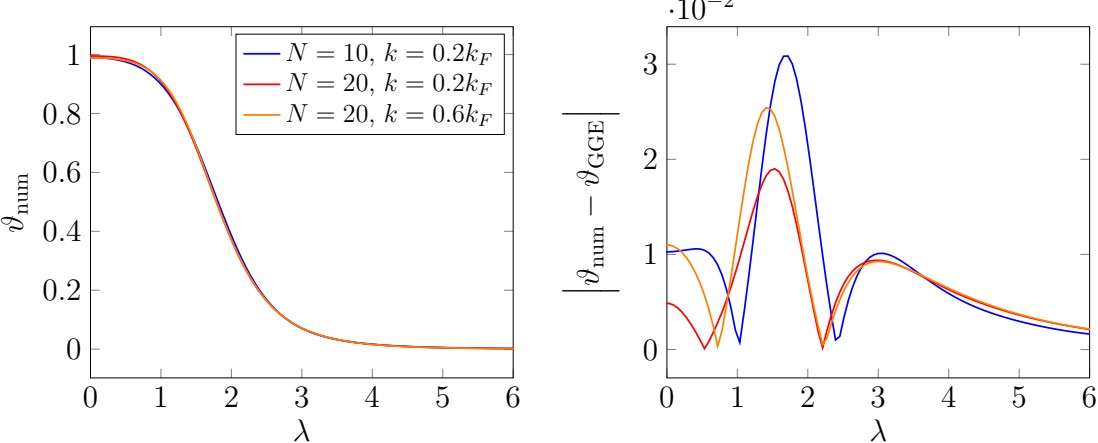

Figure 10: Left panel: Comparison between the distributions $\vartheta_{\text{num}}$ obtained from the Bragg spectra measured at different system sizes $N = 10$ and $N = 20$ and different momenta $k = 0.2\,k_F$ and $k = 0.6\,k_F$, in the quench $c = 0 \rightarrow c = 2$. Right panel: plot of the difference between the numerically computed $\vartheta_{\text{num}}$ (with the same system sizes $N$ and momenta $k$ as in the left panel) and the exact distribution $\vartheta_{\text{GGE}}$. We notice that larger system sizes $N$ and smaller momentum $k$ give a more accurate prediction for $\vartheta(\lambda)$ around $\lambda \sim 0$, while deviations in the tails remain relatively large due to the numerically truncated diagonal ensemble sum (39).

# F   A few words on the numerical evaluation of the dynamic structure factor

The ABACUS [53] software package evaluates the DSF for a fixed number $N$ of particles and finite length $L$ of the system. From this finite-size data we can infer the shape of the function in the thermodynamic limit. In Fig. 10 we check the dependence of the results on the number $N$ of particles (recall that we always work with a fixed density $n = N/L = 1$). The curves reported in the figure demonstrate that the finite-size effects are small (i.e., of order of $10^{-2}$).

The DSF $S(k, \omega)$ determined with ABACUS is a set of equally distributed discrete set of points in $k$ (quantized as $2\pi I/L$ with integer $I$) and unequally distributed points in the $\omega$ variable. In order to take the ratio in the definition of the $\Xi(k, \omega)$ we smoothed out the DSF in $\omega$ by convoluting it with a Gaussian function with width $\sigma = w\Delta E$ and $w = 0.1$ or $0.5$ and with $\Delta E$ the two-particle level spacing, and evaluating it on an equally distributed set of points in $\omega$. We have verified that this smoothening procedure does not affect the results.

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
