# Peer review of "Probing non-thermal density fluctuations in the one-dimensional Bose gas"

_SciPost Physics, doi:SciPost Phys. 3, 023 (2017)_

## Round 2 · Referee Report · Anonymous (Referee 1) · 2017-6-13

Strengths

  1. timely topic of generally high relevance in the field
  2. relation to experimental setup
  3. non-trivial advancement of field

Weaknesses

  1. presentation can be slightly improved at some points (see below)
  2. discussion of expectation values appearing in the correlation functions is not completely transparent (see below)

Report

The authors study integrable systems after a quantum quench and here focus on the stationary state at late times, which is known to be described by a GGE. Specifically they show how to relate the Lagrange multipliers appearing in the GGE to the fluctuation-dissipation ratio and thus to measurable correlation functions. They first consider general integrable systems (with one type of particles) and then show for the specific example of the Lieb-Liniger gas how their approach works in practice.

Requested changes

  1. The definition (1) is not completely transparent when it comes to the time dependence of the different ingredients. Specifically, the sub-index t refers to the expectation value in the time-evolved state at time t. Thus this quantity seems to probe the full time evolution, not just the stationary state. Also, the integration over t' also includes times before the quench (t'<0) as well as after this time (t'>t). Is the quantity defined in (1) really measurable and useful? Please clarify this point.

Furthermore, it is not clear which state the expectation values in Sec. 3.4 refer to. Is it the GE? This should be clarified before (22).

Also, how do I have to take the order of limits? For example, the text above (34) mentions S in the stationary state, which I interpret as taking t\to\infty first. The text after (34) as well as (1) give the opposite impression.

Thus the discussion around (1), in Secs. 3.4, 3.5 and 4 as well as App. A should be clarified.

Some other remarks: 2. The abstract states "The experimental measurement of this macroscopic number of temperatures remains elusive". I have two remarks here: (i) How does your statement relate to the results reported in Ref. 9? (ii) Why would you expect an integrable system and thus the GGE to be observable in experiments at all, given that integrability is for sure broken in real systems? 3. I read the statement after (7) such that one has to solve a coupled set of equations. Is this correct? Maybe clarify this in the text. 4. The wording after (31) can be improved. 5. Is it clear that the long-time limit of S is identical to the time averaged object (37)? Maybe you can justify this better? 6. In Fig. 6 the result for k=0.2 k_F is invisible. 7. Are the oscillations in Fig. 7 also an artefact of the numerical evaluation of S? The caption doesn't say anything about them.

---

## Round 2 · Referee Report · Anonymous (Referee 2) · 2017-6-16

Strengths

  1. interesting idea on a timely topic
  2. combines expertise from two different fields (TBA on one side, fluctuation-dissipation theorems on the other side)
  3. experimental relevance

Weaknesses

  1. the paper could be clearer and more concise (see suggestions for improvement below)

Report

The authors focus on a macrostate of the Lieb-Liniger model characterized by its occupation function $\vartheta (\lambda)$.
They show that the dynamical structure factor $S(k,\omega)$ at small $k$ is related to $\vartheta (\lambda)$ by a fluctuation-dissipation theorem, and they support their claim with numerical results.

This result is very appealing, and there is no doubt that it deserves to be published. I think, however, that the way
the result is presented could be improved, and made more concise. Below are some suggestions that the authors may want to take into account.

Requested changes

  1. Sections 1 and 2 are too long, in my opinion. Section 1 gives a general introduction and, in page 3, gives an overview (with words, no formulas) of what is going to be done later in the paper. Then section 2 sounds like a second introduction, with a rather long discussion of GGE that finally arrives at the fact that the authors deal with a macrostate characterized by $\vartheta (\lambda)$. The latter point is, as far as I can see, the only thing that is really needed in the paper, so the detours of section 2 (about various equivalent ways of writing a GGE) seem superfluous to me. Then section 2 ends with a second discussion (still in words, and no formulas) of what is going to be done in the paper. This is redundant.

I suggest that sections 1 and 2 be fused together, and significantly reduced.

I understand that, with their section 2, the authors want to claim that they have a "general framework" that is later "applied to the Lieb-Liniger model". But this way of presenting their result makes the paper weaker, and more confusing, because: (i) the authors do not say much about other models anyway, and their "general framework", instead of being a fluctuation-dissipation theorem valid for all integrable models (as one would expect when reading the introduction), merely consists in writing the parametrization (6) for $\vartheta (\lambda)$ (ii) the authors seem to imply that, once the Lieb-Liniger case is understood, generalization to other integrable models is presumably straightforward. I'm happy with that argument, but then there is no need for an entire section about the "general case".

  1. The reader has to wait for section 3.4 to finally get a taste of the physical reason why the occupation number should be related to the structure factor. This is too late. Section 3.4 is much more instructive than the sections 1 and 2 combined: it clearly explains the physical idea that is at the heart of the whole paper. So, it should appear much earlier; in my opinion, this section should be in the introduction.

  2. Finally, I am a bit confused by the treatment of the harmonic trap in appendix D. The authors point out two important questions that come to mind: (i) since the quantity of interest is the structure factor $S(k,\omega)$ at small $k$, it is sensitive to long-range correlations, which are more sensitive to the trap than short-range ones, and (ii) at finite $\gamma$, harmonic confinement breaks integrability.

I am not convinced by the two answers that they provide. This is not an obstacle to publication, because this is merely a non-quantitative discussion hidden in an appendix, and the paper otherwise contains many precise quantitative results. Nevertheless, maybe the authors could clarify a bit the following points: (i) the authors argue that "under actual reported experimental conditions", the difference between the DSF in a trap and in the translation-invariant case is relatively small. But is that only for a particular set of parameters? Or is a more quantitative criterion that says when this works? Then after that sentence, the authors turn to a calculation with LDA. But what is the justification for using LDA here? Is it because the macrostate in which $\left< \rho(x,t) \rho(0,0) \right>$ is evaluated is analogous to a high-temperature state, with short-range correlations only? If so, is there an estimate of the correlation length that could be compared to the scale on which the density varies, that would justify the use of LDA? (ii) The authors claim that there might be a separation of time scales $\tau_{break} \gg \tau_{rel}$. Is it possible to give an estimate of these two time scales?

---

## Round 3 · Referee Report · Anonymous (Referee 1) · 2017-8-22

Report

The authors have addressed the points raised by the referees. The manuscript seems now suitable for publication.

---

## Round 3 · Referee Report · Anonymous (Referee 2) · 2017-9-4

Report

The authors have reorganized the introductory sections of their manuscript, and I think the presentation is now much clearer. They have also answered the questions I raised. I recommend that the paper be published in its present form.

---

## Round 3 · Author Response

We thank the referees for their useful and relevant comments. Below we reply to their points and we list the changes to the manuscript.

REFEREE 1:

We very much thank Referee 1 for his/her appreciation of our work. Please find below the answers to his/her comments.

  1. We have merged Sections 1 and 2 and we think that now the paper has improved in readability.

And while we focus on the Lieb-Liniger model in this paper, our approach is completely general. It has now been shown for a wide range of models, both those with diagonal and non-diagonal scattering, that there is a simple relation between the small momentum limit of response functions and \vartheta (or its generalization), the function that encodes the GGE (Refs. 49 and 50). This is the key relation in our paper. That it exists allows us to conclude that the program proposed in this paper will work for integrable models beyond Lieb-Liniger.

  1. Following the referee's suggestion, we have included some of the content of Section 3.4 in the Introduction. Nonetheless we have left Section 3.4 in the text to allow us to give additional technical details.

  2. (i) The referee writes

“the authors argue that "under actual reported experimental conditions", the difference between the DSF in a trap and in the translation-invariant case is relatively small. But is that only for a particular set of parameters?”

We have made such statement with a chosen a set of parameters that are used in real experiments with cold atoms in optical traps, see ref [43,44]. The main point is that in real experiment the ratio \omega_{trap}/\omega is small.

The referee continues,

“Is it because the macrostate in which $\langle\rho(x,t)\rho(0,0)\rangle$ is evaluated is analogous to a high-temperature state, with short-range correlations only? If so, is there an estimate of the correlation length that could be compared to the scale on which the density varies, that would justify the use of LDA?”

Yes, the GGE state is typically a finite temperature-like state with a finite correlation length. We have commented on this in the text. However the value of this correlation length depend on the quench protocol and no general statement can be given.

(ii) We have given an estimate of these time scales in the text. They are given by the inverse of the trapping frequency and the inverse of the Fermi energy of the system respectively.

REFEREE 2:

We very much thank Referee 2 for his interest in our work. We answer his/her questions in order:

  1. We have moved Eq. (1) to Eq. (3) to only define the dynamical structure factor in the steady state, where the limit of large times has been already taken, so that the integration over time can be extended from -infinity to +infinity. More details can now be found in Section 4.

The referee writes,

“Furthermore, it is not clear which state the expectation values in Sec. 3.4 refer to. Is it the GE? This should be clarified before (22).”

The expectation value in Eq. (22) is on a generic stationary state, while the later is restricted to a thermal state and in Sec 3.5 to a GGE state. We have clarified this in the text.

The referee continues,

“Also, how do I have to take the order of limits? For example, the text above (34) mentions S in the stationary state, which I interpret as taking $t\to\infty$ first. The text after (34) as well as (1) give the opposite impression.”

The order of limits is always first t to infinity (late times after the quench) and than later t’ to infinity. We have commented on this in Sec. 4 and we have removed the ambiguity from the Introduction.

Some other remarks:

  1. (i) In Ref. 9 it was studied a particular non-equilibrium setting where a weakly interacting gas (in the semi-classical regime) was split in two. There a GGE with two effective temperature proved to be good enough to reproduce the long time limit of local observables. We here show a protocol to measure a generic number of effective temperatures that manifest themselves when a strongly interacting gas is quenched.

(ii) Numerous experimental results, including Ref. [9], have shown that even in real settings, where integrability is nominally broken, non-thermal steady states nonetheless can be observed, at least in the time scale accessible to experiments. In the past years it has been shown [4,9][26-31] that in the presence of small integrability breaking terms in the Hamiltonian, integrability is formally not broken at intermediate time scales and a GGE emerges. Then after some crossover time, the integrability-breaking perturbations start to play a role and, typically, the system relax towards a canonical Gibbs Ensemble. We have commented on this in Appendix D.

  1. “I read the statement after (7) such that one has to solve a coupled set of equations. Is this correct? Maybe clarify this in the text”

We moved this part to Sec 3.5, however the referee’s question remains still appropriate. Given functions $\beta(\lambda)$ and $\omega(\lambda)$, ${bar{\mu}}$ the chemical potential controls the total density $n=N/L$. If we want to fix the density and determine the chemical potential we have to indeed solve a set of coupled equations (13) and (15) with $\vartheta(\lambda)$ given by Eq. (32) [the old Eq. (7)]. We have added a comment in the text.

  1. We have rewritten the part after Eq. (31).

  2. For typical finite-size systems with no extensive degeneracies, the time average is equivalent to the diagonal ensemble average, which is equivalent to the GGE ensemble average, see Eq. (43) and (44). These are well established facts, see Ref. [10] for example.

  3. The two sets of data are indeed indistinguishable on the scale of the plot. We have added a comment in the caption of Fig. 6.

  4. Yes, the oscillations are also a feature of the numerics. We added this comment inside the caption of Fig 7.

---

## Editorial Decision

published